# Assessment of sST2 Behaviors to Evaluate Severity/Clinical Impact of Acute Pulmonary Embolism

**DOI:** 10.3390/ijms24054591

**Published:** 2023-02-27

**Authors:** Luigi Petramala, Antonio Concistrè, Francesca Sarlo, Silvia Baroni, Marianna Suppa, Adriana Servello, Francesco Circosta, Gioacchino Galardo, Orietta Gandini, Luca Marino, Giuseppe Cavallaro, Gino Iannucci, Claudio Letizia

**Affiliations:** 1Department of Translational and Precision Medicine, “Sapienza” University of Rome, 00185 Rome, Italy; 2Department of Clinical, Internal, Anesthesiological and Cardiovascular Sciences, “Sapienza” University of Rome, 00185 Rome, Italy; 3UOC Chimica, Biochimica e Biologia Molecolare Clinica, Fondazione Policlinico Universitario A. Gemelli I.R.C.C.S., 00185 Rome, Italy; 4Dipartimento di Scienze Biotecnologiche di Base, Cliniche Intensivologiche e Perioperatorie, Università Cattolica del Sacro Cuore, 00185 Rome, Italy; 5Emergency Medicine Unit, Department of Emergency-Acceptance, Critical Areas and Trauma, Policlinico “Umberto I”, 00185 Rome, Italy; 6Department of Molecular Medicine, “Sapienza” University of Rome, 00185 Rome, Italy; 7Department of Mechanical and Aerospace Engineering, “Sapienza” University of Rome, 00185 Rome, Italy; 8Department of Surgery Pietro Valdoni, “Sapienza” University of Rome, 00185 Rome, Italy

**Keywords:** pulmonary thromboembolism, soluble ST2, prognostic biomarker, severity risk stratification

## Abstract

Pulmonary embolism (PE) is a potentially life-threatening disorder. Beyond its usefulness in the prognostic stratification of heart failure, sST2 can represent a biomarker with high utility in several acute conditions. Our study was aimed to investigate whether sST2 can be used as a clinical marker of severity and prognostic outcome in acute PE. We enrolled 72 patients with documented PE and 38 healthy subjects; we measured the plasma concentrations of sST2 to evaluate the prognostic and severity performance of different levels of sST2 according to its association with the pulmonary embolism severity index (PESI) score and several parameters of respiratory function. PE patients had significantly higher levels of sST2 compared with healthy subjects (87.74 ± 17.1 vs. 17.1 ± 0.4 ng/mL, *p* < 0.001); we found higher PESI scores and serum lactate values in the group of patients with sST2 > 35 ng/mL compared with patients with sST2 < 35 ng/mL (138.7 ± 14.9 vs. 103.7 ± 15.1 and 2.43 ± 0.69 vs. 1.025 ± 0.05 mmol/L, respectively; *p* < 0.05). Patients with sST2 > 35 ng/mL showed higher radiological severity of PE compared with patients with sST2 < 35 ng/mL. Moreover, sST2 was the strongest parameter with a discriminative capacity for the development of acute respiratory failure and a PESI score >106 with respect to C reactive protein (CRP), creatinine, d-dimer, and serum lactate. We clearly demonstrated that sST2 significantly increased in PE and that its elevation was associated with disease severity. Therefore, sST2 may be used as a clinical marker in the evaluation of PE severity. However, further studies with larger patient populations are required to confirm these findings.

## 1. Introduction

Pulmonary embolism (PE) is a common cardiovascular emergency, associated with high mortality; due to its non-specific clinical presentations the diagnosis of PE can be missed and may evolve into acute life-threatening situations. The diagnosis of PE is based on the individual patient’s clinical probability for PE associated with laboratory and non-invasive imaging methods such as Computed Tomography Pulmonary Angiography (CTPA) [1]. There is evidence supporting the use of biomarkers to enhance the diagnosis and to predict the outcome of patients with PE. Several molecules related to both inflammatory and thrombotic PE-associated processes were investigated [2]. Soluble ST2 (sST2) is a cardiovascular injury-related biomarker, currently recognized in the stratification of mortality in heart failure as well in acute infection diseases as COVID-19 pneumonia [3,4]. The clinical impact of ST2 dates to 2005 when the IL-33 was identified as the ST2-receptor ligand [5]. Since this discovery, the axis IL-33/ST2 has been extensively studied in different pathological conditions [6]. The ST2 is now known to be present in two isoforms, the transmembrane ST2L, receptor of IL-33, and the soluble isoform sST2 whose role has been particularly considered in several diseases. In several cardiovascular pathologies, the axis IL-33/ST2L was proved to exert an anti-inflammatory response to an endothelial and/or a cardiac injury, with a protection action against hypertrophy and fibrosis development. Considering this, the soluble form sST2 was recognized to act as a decoy receptor for IL-33 with a net pro-inflammatory and pro-fibrotic effect [7]. In heart diseases, sST2 was studied in chronic and acute conditions. In acute decompensated heart failure, sST2 was proved to be a powerful predictor of short- and long-term adverse effects [8], with a cut point of 35 ng/mL of sST2 plasma concentration which separates high-risk from low-risk patients. In chronic conditions, sST2 was shown to be a strong predictor of future adverse events [9]. The role of sST2 has been considered also in other diseases, in particular for the active role in the fibrosis development in the lungs, liver, pancreas, intestine, and kidney [10,11,12]. As the main source of IL-33 and sST2, the contributions of cardiomyocytes, myofibroblasts and lung fibroblasts, and endothelial cells in response to stress or injury are well established [13,14,15]. 

In our clinical research, we measured plasma concentrations of sST2 in acute PE to evaluate the prognostic and severity performance of sST2, especially regarding its correlation with the parameters of the severity of PE (i.e., the pulmonary embolism severity index (PESI), respiratory indexes, and pulmonary vascular impairment). Hence, the aim of the present study was to evaluate the relationship between sST2 and other surrogate markers of disease severity and outcome-related parameters in patients with a newly diagnosed PE. 

## 2. Results

Anthropometric, clinical characteristics, blood gas, and echocardiographic parameters of patients are reported in Table 1. There were no differences between the two groups regarding age and sex, as well as systolic blood pressure (SBP) and diastolic blood pressure (DBP). The PE group had significantly higher prevalence of diabetes and oncologic diseases when compared with controls (6.9% vs. 3%, *p* < 0.001; 27.6% vs. 0%, *p* < 0.001, respectively). Conversely, the PE patients had significantly increased heart rate (HR) (102 ± 25.12 bpm vs. 77 ± 12 bpm, *p* = 0.03). Compared with the healthy subjects, we found significantly increased serum levels of d-dimer in the PE group (3681 ± 236 ng/mL vs. 345 ± 110 ng/mL, *p* < 0.001), as well as high-sensitivity troponin T (TNT-hs) (0.041 ± 0.0081 μg/L vs. 0.008 ± 0.002 μg/L, *p* = 0.04), fibrinogen (467.5 ± 18.8 mg/dL vs. 155 ± 43 mg/dL, *p* < 0.001), and international normalized prothrombin time (PT-INR) (1.10 ± 0.033 vs. 0.95 ± 0.3, *p* = 0.035). The analysis of biomarker levels in the whole cohort revealed that the PE patients had significantly higher levels of sST2 compared with the healthy subjects (87.74 ± 17.1 ng/mL vs. 17.1 ± 0.4 ng/mL, *p* < 0.001). We did not find significant differences of sST2 behaviors regarding the presence of diabetes (diabetic group, 114.8 ± 85.1 ng/mL vs. non-diabetic group, 84.5 ± 17 ng/mL; *p* = 0.58) or oncologic diseases (oncologic group, 94.18 ± 27 ng/mL vs. non-oncologic group, 85.68 ± 21 ng/mL; *p* = 0.91). The blood gas data showed a lower ratio of arterial oxygen partial pressure to fractional inspired oxygen (P/F ratio) in the PE patients (273.22 ± 101.7 vs. 410 ± 30.5, *p* = 0.02) and increased values of serum lactate (1.76 ± 0.37 mmol/L vs. 0.8 ± 0.02 mmol/L, *p* < 0.001) compared with controls. Moreover, we found higher levels of TNT-hs in the EP group with respect to the controls (0.008 ± 0.002 μg/L vs. 0.041 ± 0.0081 μg/L, *p* = 0.04). The echocardiographic data highlighted a higher prevalence of left ventricular hypertrophy in the PE patients than in the controls (28% vs. 5%, *p* = 0.02), as well as higher right ventricle diameter at the base (RVd) (39.54 ± 4.71 mm vs. 32.2 ± 5.5 mm, *p* = 0.012) and estimated right ventricular systolic pressure (RVSP) (34.36 ± 7.1 mmHg vs. 20.12 ± 8.34 mmHg, *p* = 0.03) and lower right ventricle free wall longitudinal myocardial velocity (RVFW S’) (11.37 ± 2.88 cm/sec vs. 12.6 ± 1.23 cm/sec, *p* = 0.04) and right ventricular fractional area change (RVFAC) (36.2 ± 8.5% vs. 42.3 ± 5.12%, *p* = 0.02), respectively. To distinguish overall patients in relation to a cutoff value of 35 ng/mL of sST2, where the cutoff was recognized to be associated with a worse prognosis in heart failure, we found significantly higher PESI scores and serum lactate values in the group of patients with sST2 values more than 35 ng/mL compared with patients with sST2 values less than 35 ng/mL (138.7 ± 14.9 vs. 103.7 ± 15.1 and 2.43 ± 0.69 mmol/L vs. 1.025 ± 0.05 mmol/L, respectively, *p* = 0.05) (Table 1). Moreover, the patients with sST2 > 35 ng/mL showed higher CRP levels (12.65 ± 2.59 mg/dL) than the group with sST2 < 35 ng/mL (4.81 ± 1.52 mg/dL; *p* < 0.02). Interestingly, we found higher radiological severity of PE in the patients with sST2 > 35 ng/mL compared with the patients with sST2 < 35 ng/mL; in particular, we observed 33% with massive PE, 47% with submassive PE, and 20% with segmental PE in patients with higher levels of sST2. Otherwise, patients with sST2 < 35 ng/mL had submassive PE in 50% of cases and segmental PE in the other 50% of cases (Figure 1). Moreover, in Figure 2 we reported the behavior of sST2 in the control group (17.1 ± 0.4 ng/mL) and in the PE group distinguished according to the PESI score, showing that the severity of PE evaluated by the PESI score was strictly associated with plasma sST2 levels: high (class VI–V, PESI score > 106), 111.7 ± 13.6 ng/mL; intermediate (class III, PESI score 86–105), 42.8 ± 16.3 ng/mL; and low (class I–II, PESI score < 85), 27.2 ± 2.1 ng/mL.

Table 2 shows the correlation matrix for selected continuous variables analyzed. We observed a positive correlation between sST2 and the PESI score (r = 0.434, *p* = 0.036) and a negative correlation between sST2 and the P/F ratio (r = −0.597, *p* = 0.015); moreover, we found a significative correlation between sST2 and a higher amount of oxygen flow supplied (r = 0.554, *p* = 0.003). Among other parameters, we found a negative correlation between serum lactate and pCO_2_ (r = −467, *p* < 0.05).

In order to determine the diagnostic role of sST2 in terms of prediction of acute respiratory failure (P/F < 300), we performed a receiver operating characteristic curve (ROC curve) (Figure 3). This analysis showed that the strongest parameter with a significant discriminative capacity for the development of acute respiratory failure (P/F < 300) was sST2, with an area under the curve (AUC) of 0.861 (*p* < 0.001), with respect to PESI score (AUC = 0.766, *p* = 0.005), CRP (AUC = 0.680, *p* = 0.054), creatinine (AUC = 0.572, *p* = 0.440), d-dimer (AUC = 0.422, *p* = 0.407), and serum lactate (AUC = 0.564, *p* = 0.491). We performed another ROC curve to determine the diagnostic role of sST2 in terms of prediction of a severe PESI score (>106). As shown, sST2 was the strongest parameter with a significant discriminative capacity for worse PE prognosis, with AUC of 0.800 (*p* = 0.007), with respect to CRP (AUC = 0.775, *p* = 0.103), creatinine (AUC = 0.486, *p* = 0.113), d-dimer (AUC = 0.632, *p* = 0.114), and serum lactate (AUC = 0.816, *p* = 0.077) (Figure 3).

## 3. Discussion

PE induces the worsening of the respiratory capacity through both the alteration of gas exchanges and changes of the hemodynamic conditions, by mechanical and biochemical mechanisms. In fact, severe PE can result in increased pulmonary artery pressure (PAP), acute pressure overload, subsequent right ventricular (RV) failure, and impaired RV filling, until the desynchronization of the both ventricles [16]. Several cytokines (i.e., thromboxane A2, serotonin, endothelin-1, vasopressin, and copeptin), neurohormonal activation, and myocardial inflammatory cells are responsible for significant arterial changes, contributing to increased pulmonary vascular resistance (PVR), worsening RV failure, and myocardial damage [17,18,19]. These conditions can evolve into reduced left cardiac output, contributing to systemic hypotension and systemic haemodynamic instability [20]. Considering this, the finding of elevated circulating levels of biomarkers of vascular or myocardial injury is representative for adverse outcomes of acute and severe PE [21]. Moreover, all these haemodynamic disturbances favor ventilation/perfusion mismatch, alveolar hemorrhage, pleural effusion, and subsequent worsening of respiratory failure [22]. Because of this, the clinical management requires identification immediately in life-threatening situations as overt haemodynamic instability or RV dysfunction through the identification of biochemical prognostic markers. sST2 is the soluble form of protein expressed through several cells (basophiles, CD4 lymphocytes, eosinophils, macrophages, fibroblast, endothelial cells, and keratinocytes), representing the natural ligand of interleukin-33 (IL-33). It can be produced spontaneously in the lung, kidney, heart, and the small intestine, with the role of IL-33 soluble receptor, blocking the effects in the target cells, inhibiting the activation of Th2 cell response and the release of anti-inflammatory cytokines (IL-4, IL-5, IL-10, and IL-13), and polarizing the Th1 response, which results in the activation and release of inflammatory cytokines (TNF-α) and inflammation [4]. Initially evaluated in relation to neoplastic, inflammatory conditions, fibroproliferative diseases, autoimmune diseases, and systemic infections, sST2 is widely recognized as a strong biomarker with high prognostic power in cardiovascular diseases, extremely useful in cardiac remodeling and ischemic heart disease [5]. Considering this, in relation to pro-B-type natriuretic peptide (pro-BNP), sST2 showed similarly high diagnostic power for heart failure, but it additionally possesses a stronger ability to predict fatal events (in-hospital and 1-month mortality rates). In relation to respiratory insufficiency not directly due to heart diseases, an important multicenter randomized trial showed that sST2 was significantly associated with the inflammatory cascade of acute respiratory distress syndrome; particularly, patients with higher values during ventilatory treatment showed decreased probability of extubation and higher need for reintubation [23]. First, in the literature, our study shows a significant correlation between severity and the need for oxygen support in pulmonary embolism with circulating levels of sST2. In fact, in our casuistry of patients with acute PE, beyond the expected increased mean heart rate and d-dimer values, we interestingly found a significant increase in the circulating behaviors of sST2, negatively related to the P/F ratio, and positively related to the need of higher FiO_2_, serum lactate, and PESI score. Due to the complex physiopathological alterations secondary to mechanical, haemodynamic, and cytokine changes, the reduction of oxygenation is one of the typical features of PE, especially resistant to oxygen supplementation [24]. In a relevant study with long-term follow-up, Wang et al. found that a P/F ratio <265 at the diagnosis showed an odds ratio (OR) of 6.3 for predicting in-hospital mortality, a stronger marker than others such as history of cancers (OR 4.3) or severity of PE stratification (OR 4.2) [25]. Considering this, we have found the greater ability of sST2 in predicting significant respiratory insufficiency compared with other well-recognized parameters such as lactates, PESI index, and circulating levels of D-Dimer. Lactate is generated in the presence of an imbalance between tissue oxygen supply and demand, beginning a relevant and significant marker of haemodynamic compromise. Vanni et al. showed in PE patients that increased lactate level (>2 mmol/L) was related to higher 30-day mortality (OR 11.7) and short-term clinical deterioration (shock, mechanical ventilation, or cardiopulmonary resuscitation), with an OR of 17.9 with hemodynamic complications, RV dysfunction, or elevated troponins [26]. Moreover, recent meta-analysis showed the sensitivity (67%), specificity (73%), positive likelihood ratio (2.5), and negative likelihood ratio (0.45) of lactate for predicting overall mortality in acute PE patients, with the AUC of the ROC curve of 0.76, confirming that its evaluation can be routinely measured in risk stratification. In the management of PE patients without haemodynamic stability, the identification of parameters and scores useful to risk stratification and early management is surely recommended. In relation to this, the combination of clinical, imaging, and laboratory parameters showed interesting data on prognostic purposes (i.e., Bova score and FAST score) [6,27,28,29]. To date, the association of a positive troponin test and RD dysfunction detected by echocardiogram or CTPA has been proven to be a useful method for early therapeutic treatment in acute PE [30]. As a result, it is useful to evaluate other possible comorbidities and aggravating conditions that are able to influence overall mortality risk and early outcomes. Regarding these scores of severity and risk, the PESI score and its simplified version (sPESI) are the most extensively validated scores to identify 30-day mortality, although not prospectively used to guide therapeutic management of low-risk PE patients [31,32,33,34,35]. Whereas class I–II PESI scores are a predictor of low risk (overall 30-day mortality rate 0.5–1%), in the high-risk group for PESI (>106), the overall mortality rate is 20–30% [32,36,37]. Relating to this, our study considering different parameters showed that circulating levels of sST2 were more predictive than the D-dimer assessment in predicting a PESI score >106. In addition to its fundamental role in the diagnosis of PE with a sensitivity of 83% and a specificity of 96%, computed tomographic pulmonary angiography is also useful in assessing the extent of pulmonary vascular damage, allowing adequate visualization of the pulmonary arteries down to the subsegmental level, as well as identifying chronic thromboembolic pulmonary hypertension (CTEPH), a potentially fatal late sequela of PE [38,39,40]. On acute low-risk PE, a recent meta-analysis showed computed tomography-assessed RV enlargement was present in 34% of subjects, with higher 30-day mortality (OR 2.6) [41]. In our study, we observed that the group of patients with sST2 values >35 ng/mL (the threshold recognized as predictive of acute cardiac complications [42]) showed the most extended embolic occlusion of pulmonary arteries (20% for massive PE and 47% submassive PE), compared with the group with values less than 35 ng/mL (50% submassive PE, no massive PE); these findings confirmed the meaningful correlation between circulating levels of sST2 and the extension of damage in the pulmonary arterial circulation, suggesting the useful role of ST2 evaluation in highlighting the grade of vascular pulmonary involvement. The extension of vascular damage as well as the functional damage were also confirmed by higher values of Lactates, D-Dimers, and PESI scores in patients with sST2 values >35 ng/mL. To date, sST2 is generally considered useful in cardiovascular diseases, with a stronger predictive ability of adverse outcome across a wide spectrum of disease, especially in heart failure, as well as in acute dyspnea and pulmonary diseases [43,44]. In a Swedish study on patients with acute hypercapnic respiratory failure treated with non-invasive positive pressure ventilation (NPPV), sST2 was an independent predictor of both short-term and long-term mortality during the follow-up [44]. 

sST2 is thought to play a role in conditions of increased stress of the vascular wall as well as cardiac overload, especially those characterized by high grades of vascular inflammation or atherosclerosis [45]. In consideration of its specific characteristic of vascular stress biomarker, with a significant key role as a regulator of immunity and inflammation, sST2 can be considered a potential prognostic marker rather than a specific indicator of disease. The role of sST2 is acknowledged as a biomarker of myocardial remodeling, interstitial fibrosis, and vascular dysfunction, due to impairment of myocardial cells, fibroblasts, and endothelial cells, characterized by shear stress, proinflammatory milieu, and oxidative stress [46]. Therefore, in acute PE, increased wall stress, cardiac overload, and respiratory failure can induce the significant increase in circulating sST2 levels, becoming a marker of the extent of vascular involvement as well as cardiac overload, and as a result of the need for oxygen support (Figure 4).

Our study was limited by the fact is that we evaluated the circulating levels of sST2 only at the diagnosis of PE, not re-evaluating it also at the end of treatment and hospitalization; furthermore, we currently have no information regarding the medium- to long-term outcomes of patients in relation to the circulating levels of sST2. Further studies, conducted on larger samples, are needed to explain the possible prognostic significance of this biomarker.

Finally, the rapid stratification of the global damage in acute PE and the predictor of ventilatory failure are critical to prevent complications and improve adequate treatment in PE patients; due to its biological characteristics, the sST2 can represent a useful biomarker for these purposes.

## 4. Materials and Methods

This observational cohort study included 72 consecutive patients (37 males and 35 females; mean age: 68 ± 2.8 years) admitted to the Emergency Medicine Department, Policlinico Umberto I Hospital-Rome, between 1 October 2021 and 31 July 2022 who had a diagnosis of PE at hospital admission. PE was diagnosed by spiral CTPA in all patients with suspected PE by clinical and laboratory evaluation. Thirty-eight healthy subjects (17 males and 21 females; mean age: 58 ± 6.4 years) with no previous history of disease and normal physical examination findings were recruited into the control group and included in the study. Data were collected on the clinical and laboratory findings of the patients at the diagnosis including the complete blood count (CBC), lactate dehydrogenase (LDH), CPR, ferritin, d-dimer, TNT-hs, prothrombin time (PT), activated partial thromboplastin time (aPTT), creatine phosphokinase (CPK), electrolytes, liver enzymes, arterial blood gas values, and the presence of underlying diseases or predisposing factors for PE. In addition to these parameters, the sST2 serum concentration was measured in all patients. All consecutive patients with suspicious clinical symptoms associated with PE admitted to our Emergency Medicine Department were evaluated as prospective study subjects. Blood samples were taken from the brachial vein using vacutainer tubes without anticoagulants. All patients underwent transthoracic echocardiography to evaluate main echo-color doppler parameters. The PESI score was used for clinical scoring to stratify the PE enrolled patients into five different risk classes [47]. Based on the severity of the arterial filling defects detected by CTPA, three subgroups were defined (massive, submassive, and segmental PE). Overall, the subjects of the present study cohort were classified in two groups, according to the sST2 values. In particular, the groups were distinguished using the cutoff of 35 ng/mL known to be associated with a worse prognosis in heart failure [48]. The exclusion criteria of the patient group were as follows: acute ischemic disease such as acute cerebrovascular disease, acute coronary syndrome, acute peripheral arterial occlusion, or acute intestinal ischemia; advanced heart failure, chronic kidney disease, liver cirrhosis, or inflammatory diseases; or refusal to participate in the study.

As previously validated [49] we used the sST2 SEQUENT-IA™ kit, a turbidimetric immunoassay (Critical Diagnostics, San Diego, CA, USA) implemented in the ADVIA Chemistry XPT System (Siemens Healthcare Diagnostics, Erlangen, Germany) according to the procedure provided by the manufacturer. The analytical performance of the assay was evaluated according to the Clinical and Laboratory Standards Institute (CLSI) EP15-A3 guidelines [50].

We designed this study in order to evaluate the prognostic and severity performance of sST2 in acute PE, respecting other parameters such as the PESI score and respiratory index. 

We performed statistical analysis using SPSS 28.0 for Mac OS (SPSS, Chicago, IL, USA). The data were expressed as means ± standard deviation (SD). A power analysis was performed to determine the sample size; alpha = 0.05, and the power of the test was calculated as 0.7. Differences between means were assessed by the Student’s *t* test or the Mann–Whitney *U* test in non-normally distributed data for two-sample comparison or by one-way analysis of variance (ANOVA) applying the Fisher least significant difference post hoc test for multiple comparisons. *Χ*^2^ statistics were used to assess the differences between the categorical variables. The relationships between continuous variables were assessed by calculating the Pearson correlation coefficient or the Spearman rank correlation coefficient when appropriate. We compared the predictive performance of sST2, PESI, CRP, creatinine, d-dimer, and serum lactate [area under the curve (AUC)] as continuous variables using receiver operating characteristic (ROC) curves and by calculating the AUCs in the detecting values of P/F < 300 and PESI > 106. *p* values less than 0.05 were taken as statistically significant.

## Figures and Tables

**Figure 1 ijms-24-04591-f001:**
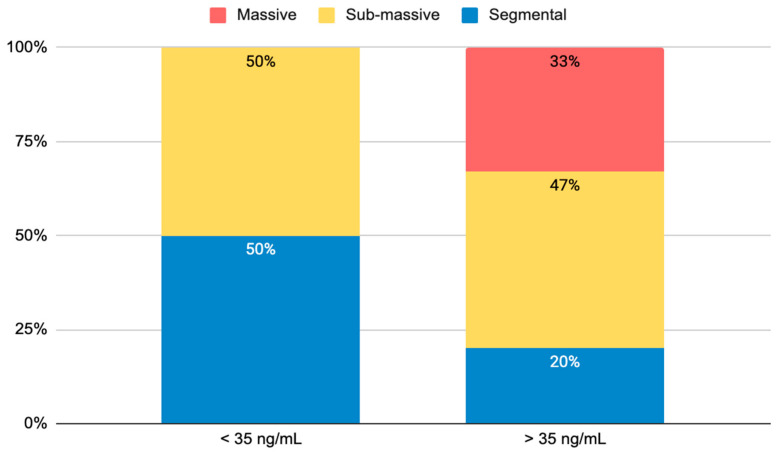
Radiological severity of pulmonary embolism dividing patients according to the sST2 cutoff of 35 ng/mL.

**Figure 2 ijms-24-04591-f002:**
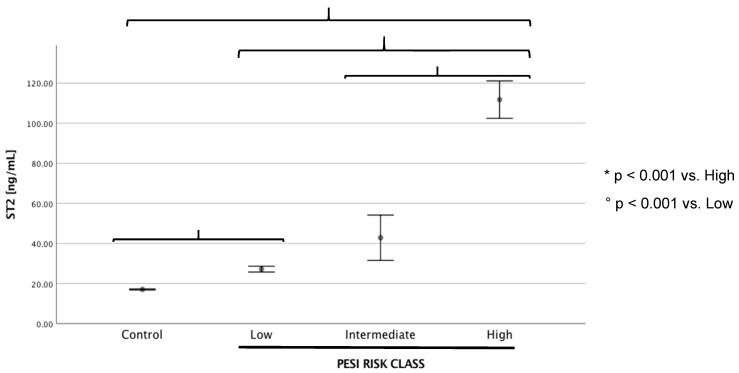
sST2 behaviors (mean ± sd) in the control group and in the PE group as distinguished according to PESI score: low (class I–II, < 85), intermediate (class III, 86–105), and high (class IV–V, >106).* refers to comparisons with high PESI risk class, ° refers to comparisons with low PESI risk class.

**Figure 3 ijms-24-04591-f003:**
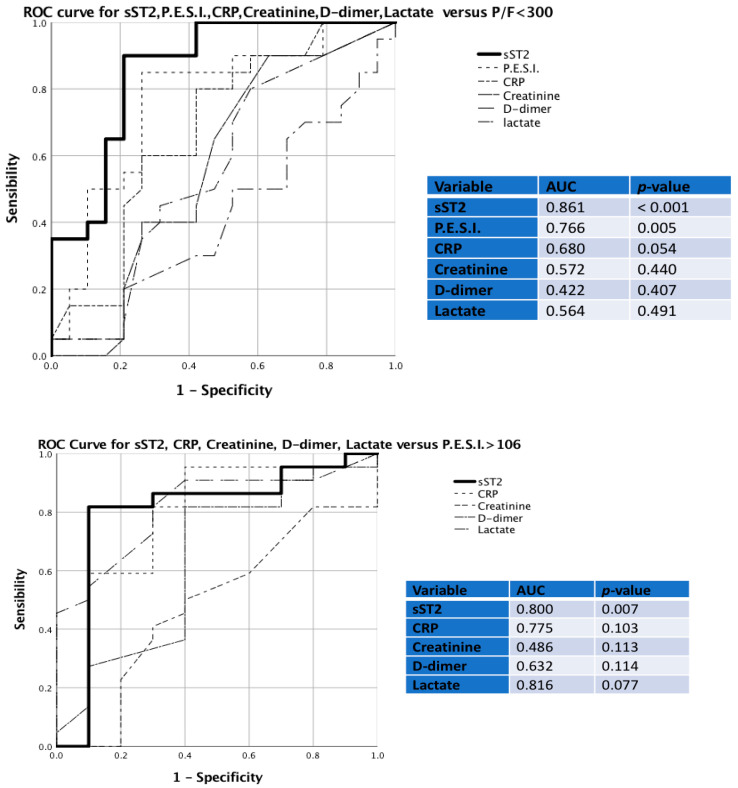
Receiver operating characteristic (ROC) curves in PE patients. Top panel: sST2, PESI score, CRP, creatinine, D-dimer, and lactate for distinguishing the development of acute respiratory failure (P/F < 300). Bottom panel: sST2, PESI score, CRP, creatinine, D-dimer, and lactate for distinguishing a worse prognosis (PESI score > 106).

**Figure 4 ijms-24-04591-f004:**
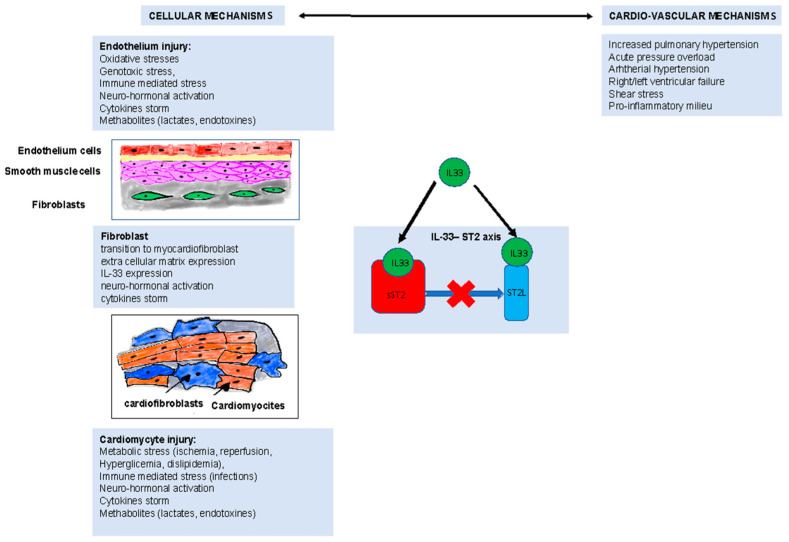
Summary of mechanisms underlying the fibrogenesis process. Cellular mechanisms are triggered by damage to the endothelium, cardiomyocyte, and fibroblasts, leading to altered signaling of the IL-33/ST2L axis and promoting pro-fibrotic processes.

**Table 1 ijms-24-04591-t001:** Anthropometric, biochemical, blood gas, and echocardiographic parameters in overall patients enrolled and in PE patients distinguished on sST2 behaviors.

	Controls (*n* = 38)	PE (*n* = 72)	*p*-Value	PE < 35 ng/mL (*n* = 34)	PE > 35 ng/mL (*n* = 38)	*p*-Value
Anthropometric parameters						
Age (years)	58 ± 6.4	68 ± 2.8	0.71	67 ± 4.7	70 ± 24.1	0.64
Males (%)	45	51.4	0.9	50	52.6	0.88
Hypertension (%)	30	43	0.57	41.2	44.7	0.83
Diabetes (%)	3	6.9	<0.001	8.8	5.2	0.12
Heart failure/cardiomyopathy (%)	3	1.08	0.567	0.0	2.6	0.07
Obesity (%)	17	41.6	0.35	47	36.8	0.43
Oncologic (%)	0	27.6	<0.001	17.6	34.72	0.18
SBP (mmHg)	127 ± 11.8	134 ± 11.8	0.08	134 ± 3.9	132 ± 9.8	0.89
DBP (mmHg)	62 ± 8.8	76 ± 15.49	0.54	79 ± 4.6	75 ± 4.3	0.51
HR (bpm)	77 ± 12	102 ± 25.12	0.03	100 ± 8.5	104 ± 6.0	0.74
RR (bpm)	17 ± 6	19.12 ± 5.62	0.76	18.3± 5.8	20.1 ± 4.2	0.44
Laboratory parameters						
D-dimer (n.r. 0–550 ng/mL)	345 ± 110	3681 ± 236	<0.001	3164 ± 472	4030 ± 196	0.08
Creatinine (n.r. 0.1-1.2 mg/dL)	0.8 ± 0.2	1.06 ± 0.16	0.075	1.23 ± 0.34	0.94 ± 0.08	0.36
TNT-hs (n.r. < 0.0014 μg/L)	0.008 ± 0.002	0.041 ± 0.0081	0.04	0.040 ± 0.010	0.045 ± 0.012	0.78
Fibrinogen (n.r. 200–400 mg/dL)	155 ± 43	467.5 ± 18.8	<0.001	433.9 ± 33.9	491 ± 22.5	0.15
C-reactive protein (n.r. < 0.5 mg/dL)	1.5 ± 0.92	9.07 ± 5.64	0.065	4.81 ± 1.52	12.65 ± 2.59	0.02
PT-INR (n.r. 0.8–1.2)	0.95 ± 0.3	1.10 ± 0.033	0.035	1.05 ± 0.04	1.16 ± 0.05	0.12
aPTT (n.r. 0.8–1.2)	0.64 ± 0.28	0.92 ± 0.031	0.2	0.91 ± 0.042	0.92 ± 0.051	0.88
sST2 (ng/mL)	17.1 ± 0.4	87.74 ± 17.1	<0.001	24 ± 1.71	138 ± 23.8	<0.001
PESI score	35 ± 18.3	117.5 ± 10.6	<0.05	103.7 ±15.1	138.7 ± 14.9	0.05
Blood gas parameters						
pH	7.40 ± 0.23	7.47 ± 0.14	0.8	7.44 ± 0.12	7.48 ± 0.21	0.038
pO_2_ (mmHg)	91.7 ± 5.1	89.7 ± 3.9	0.2	88.1 ± 5.14	91.6 ± 6.08	0.011
pCO_2_ (mmHg)	39.2 ± 2.3	35.55 ± 1.42	0.025	36.6 ± 5.7	33.9 ± 2.08	0.37
FiO_2_ (%)	21	34.86 ± 2.32	0.02	29.5 ± 2.80	40.53 ± 3.27	0.04
SpO_2_ (%)	96.7 ± 1.3	95.2 ± 1.01	0.15	96.4 ± 2.3	93.4 ± 2.5	0.28
P/F ratio	410 ± 30.5	273.2 ± 101.7	0.02	301.43 ± 39.48	228.67 ± 30.19	0.07
Lactate (mmol/L)	0.8 ± 0.02	1.76 ± 0.37	<0.001	1.025 ± 0.05	2.43 ± 0.69	0.05
Echocardiographic parameters						
LVIDD (mm)	48.3 ± 5.2	49.9 ± 4.7	0.231	50.4 ± 4.9	49.2 ± 4.4	0.65
IVSd (mm)	9.41 ± 0.42	10.36 ± 1.12	0.032	9.78 ± 0.99	10.8 ± 1.33	0.43
LVPWd (mm)	9.11 ± 0.92	9.81 ± 1.25	0.348	9.62 ± 1.23	10.1 ± 1.25	0.22
LAVI (mL/m^2^)	24.13 ± 3.12	27.8 ± 6.38	0.094	26.7 ± 5.88	28.7 ± 6.63	0.31
LVM (g)	179.2 ± 26.2	189.27 ± 37.1	0.128	186.7 ± 29.89	195.2 ± 39.82	0.04
LVMi (g/m^2^)	90.2 ± 12.3	99.32 ± 17.56	0.099	94.75 ± 15.67	102.28 ± 19.7	0.035
LVEF (%)	58.3 ± 5.2	54.27 ± 7.86	0.134	55.14 ± 9.32	52.27 ± 6.34	0.12
LVH (%)	5	28	0.02	42	58	0.11
RVd (mm)	32.2 ± 5.5	39.54 ± 4.71	0.012	37.54 ± 4.71	40.72 ± 3.12	0.06
TAPSE (mm)	22.21± 3.11	18.72 ± 3.13	0.002	19.88 ± 4.12	17.12 ± 2.92	0.05
RVFW S’ (cm/sec)	12.6 ± 1.23	11.37 ± 2.88	0.04	11.88 ± 3.79	10.97 ± 2.2	0.12
RVFAC (%)	42.3 ± 5.12	36.2 ± 8.5	0.02	39.5 ± 9.9	34.7 ± 7.8	0.021
RVSP (mmHg)	20.12 ± 8.34	34.36 ± 7.1	0.003	33.6 ± 3.5	34.94 ± 7.9	0.38

SBP: systolic blood pressure; DBP: diastolic blood pressure; HR: heart rate; RR: respiratory rate; TNT-hs: high-sensitivity troponin T; PT-INR: prothrombin time—international normalized ratio; aPTT: activated partial thromboplastin time; PESI: Pulmonary Embolism Severity Index; LVIDD: left ventricular internal dimension in diastole; IVSd: end-diastolic interventricular septum; LVPWd: end-diastolic left ventricular posterior wall; LAVI: left atrial volume indexed for body surface area; LVM: left ventricle mass; LVMI: left ventricle mass indexed for body surface area; LVH: left ventricular hypertrophy; RVd: right ventricle diameter at the base; RVFW S’: right ventricle free wall longitudinal myocardial velocity (S’); RVFAC: right ventricular fractional area change; RVSP: estimated right ventricular systolic pressure.

**Table 2 ijms-24-04591-t002:** Simple correlation matrix for selected continuous variables analyzed in PE patients.

Pearson Correlation Coefficient *p*-Value	Age	FBG	PT-INR	aPTT	d-DIM	TNT-hs	Creat	CRP	PESI	sST2	FiO_2_	pO_2_	pH	pCO_2_	Lactate	P/F
Age		0.160	0.070	0.068	0.108	0.006	0.221	**0.452**	**0.444**	0.160	−0.045	−0.134	−0.034	0.084	0.032	0.23
		0.415	0.725	0.732	0.616	0.980	0.268	0.018	0.050	0.425	0.817	0.554	0.888	0.725	0.898	0.92
FBG			−0.002	0.239	−0.263	−0.104	−0.080	**0.508**	0.086	0.163	−0.001	0.096	−0.124	-.124	0.026	−0.095
			0.993	0.221	0.215	0.653	0.692	0.007	0.720	0.425	0.994	0.671	0.602	0.602	0.916	0.70
PT-INR				**0.595**	0.235	0.042	0.175	0.316	0.206	0.427	**0.380**	−0.079	−0.321	−0.092	**0.643**	**−0.423**
				0.001	0.268	0.856	0.382	0.108	0.384	0.029	0.046	0.725	0.168	0.700	0.003	0.05
aPTT					0.013	−0.292	0.078	0.313	0.090	0.349	0.149	−0.193	−0.406	0.216	0.315	−0.479
					0.953	0.200	0.699	0.111	0.706	0.081	0.448	0.389	0.076	0.360	0.190	0.07
d-DIM						0.295	−0.025	−0.202	0.338	0.116	0.171	0.031	0.189	−0.047	0.329	0.011
						0.207	0.908	0.356	0.170	0.598	0.424	0.899	0.467	0.859	0.214	0.9
TNT-hs							0.114	−0.203	0.009	−0.126	0.144	0.716	0.050	−0.570	0.152	0.303
							0.623	0.376	0.971	0.595	0.534	0.236	0.850	0.017	0.574	0.25
Creatinine								0.070	−0.124	−0.075	0.083	−0.324	0.001	0.255	−0.061	−0.032
								0.734	0.603	0.717	0.680	0.152	0.996	0.292	0.810	0.90
CRP									0.063	0.304	0.277	0.003	−0.155	0.002	0.199	−0.295
									0.793	0.139	0.163	0.990	0.514	0.992	0.414	0.23
PESI										**0.434**	0.520	0.009	0.107	0.133	0.150	−0.28
										0.036	0.019	0.974	0.693	0.623	0.594	0.33
sST2											**0.554**	−0.171	0.104	−0.127	0.441	**−0.596**
											0.003	0.471	0.682	0.615	0.076	0.015
FiO_2_												−0.132	−0.030	−0.092	0.367	**−0.906**
													0.901	0.699	0.123	0.001
pO_2_													−0.141	−0.227	−0.199	0.39
													0.553	0.336	0.413	0.1
pH														0.076	0.152	0.134
														0.749	0.535	0.59
pCO_2_															**−0** **.467**	0.09
															0.044	0.72
Lactate																−0.35
																016
P/F																

FBG: fibrinogen; d-DIM: d-dimers; TNT-hs: high-sensitivity troponin T; Creat: creatinine; CRP: C reactive protein; FiO_2_: fraction of inspired oxygen.

## Data Availability

Not applicable.

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
