# Peer review of "Assessment of sST2 Behaviors to Evaluate Severity/Clinical Impact of Acute Pulmonary Embolism"

_ijms, 2023, doi:10.3390/ijms24054591_

Round 1
Reviewer 1 Report
The study of new biomarkers for assessing the severity of pulmonary embolism is very modern and interesting. But:
1. In this article only the PESI score was taken into account for determining the severity of PE patient and not all the components that define the risk of patients with pulmonary embolism: haemodynamic instability, clinical parameters of PE severity and/ or comorbidity (PESI class III–V or sPESI ≥1), RV dysfunction on TTE or CTPA and changes in cardiac troponin levels. The studied group was not analyzed according to the risk of early (in-hospital or 30 day) death, but but only from the perspective of a group of patients with pulmonary embolism vs a healthy control group. That is why the statement on the relationship between increased levels of sST2 and more severe changes on CT is not clear if it refers only to the group of patients with PE (!) especially since the figures do not specify the number of patients in each category included in the respective analysis.
2. Another limitation that must be remedied is related to the emphasis placed only on the respiratory component of PE when what is important in the pathophysiology and therefore also in the evolution of this disease is the right ventricle (RV) dysfunction.
I recommend the authors to take the following aspects into consideration:
to redo the analysis taking into account the risk levels (low, intermediate and high) of the patients with PE (a small group, however), to use the control group only in the assessment of the trend of this biomarker, to add data regarding RV dysfunction in the group with PE.
Reviewer 2 Report
Petramala et al. investigated the sST2 in patients with acute PE. sST2 was higher in patients with acute PE compared with healthy control. sST2 was associated with diseases severity. This study is interested; however, some concerns are included.
1. PE induced by oncology was about 27%. Did oncologic factors affect the sST2 level?
2. Acute PE sometimes causes right heart failure. sST2 is a cardiovascular injury-related biomarker. Did authors examine right heart function? Authors should show the findings of echocardiography.
3. Acute PE sometimes causes pulmonary hypertension. Did authors examine the association between sST2 and pulmonary artery pressure, pulmonary vascular resistance, and estimated right ventricular systolic pressure from echocardiography?
4. What is the usefulness for diagnostic tool in combination of sST2 and D-dimers?
5. Authors should show the risk of PE (obesity, drugs and so on) in Table1.
Reviewer 3 Report
The manuscript has evaluated the potential role of sST2 as a biomarker in pulmonary embolism. The conclusion is supported by the results provided. However, the manuscript should be thoroughly checked for the English language and grammar (please see yellow highlights for example). In multiple places, very long sentences have been used, please modify them. Further, the authors have not mentioned whether these patients were ventilated. This is important because mechanical ventilation increases sST2 levels. Please discuss this aspect in the discussion section. Additionally, cardiovascular disease also increases sST2, was there any history of heart failure or cardiomyopathy in these patients?

Round 2
Reviewer 1 Report
Please, replace this Table 1. Anthropometric, biochemical, blood gas parameters and echocardiografic parameters in control, PE with sST2 values less or greater than 35 ng/mL groups
Author Response
Thank you very much for giving us the chance to revise the manuscript (ID ijms-2132154 entitled “Assessment of sST2 behaviors to evaluate severity/clinical impact of acute pulmonary embolism”). We are grateful to the editor and reviewer for their comments and suggestions. We have revised the manuscript as suggested.
Enclosed are the point-by-point responses to the reviewer comments; the manuscript has been exactly revised and the changes highlighted in yellow.
Now, I hope this version of the manuscript can be acceptable for publication in “International Journal of Molecular Sciences” in this version.
Again, appreciate your attention and efforts, I look forward to hearing from you soon.
Reviewer Comments:
REFEREE 1
- Please, replace this Table 1. Anthropometric, biochemical, blood gas parameters and echocardiografic parameters in control, PE with sST2 values less or greater than 35 ng/mL groups
Answer: thank you for consideration and opinion; we have modified Table 1 as suggested, removing table 2 from paper.
Reviewer 2 Report
I am satisfied with authors' responses.
Author Response
Thank you for your opinion.
Reviewer 3 Report
None
Author Response
Thank you for your opinion.